# Cascading Operators in CAT(0) Spaces

Helga Fetter Nathansky [1,\*,†] and Jeimer Villada Bedoya [2,†]

1. Centro de Investigación en Matemáticas (CIMAT), 36240 Guanajuato, Mexico
2. Institute of Mathematics, Marie Curie Skłodowska University, 20-031 Lublin, Poland; villadabedoyaj@office.umcs.pl
\* Correspondence: fetter@cimat.mx
† These authors contributed equally to this work.

**Abstract:** In this work, we introduce the notion of cascading non-expansive mappings in the setting of CAT(0) spaces. This family of mappings properly contains the non-expansive maps, but it differs from other generalizations of this class of maps. Considering the concept of Δ-convergence in metric spaces, we prove a principle of demiclosedness for this type of mappings and a Δ-convergence theorem for a Mann iteration process defined using cascading operators.

**Keywords:** cascading non-expansive mappings; CAT(0) space; fixed point property; Mann iteration

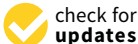



## 1. Introduction

In [1], Lennard et al. introduced a class of nonlinear operators in Banach spaces called *cascading non-expansive mappings* which generalizes the non-expansive mappings. These mappings arise naturally in the setting of Banach spaces which contain an isomorphic copy of $\ell_1$ or $c_0$ and some important concepts like reflexivity [1] and weak compactness [2] have been characterized in terms of the fixed point property for this family of operators.

Although these mappings were introduced in the framework of Banach spaces, its definition depends fundamentally on three properties of the space, the metric, the completeness, and the concept of convexity, so it makes sense to define cascading operators in metric spaces where there is a notion of convexity.

A nonlinear setting which is natural to extend the concept of cascading operator is that of uniquely geodesic metric spaces, since in these spaces the notion of geodesic allows a definition of convex sets. In relation to fixed point problems, it has been specially fruitful to consider the subclass of CAT(0) spaces, which possesses a metric structure that is similar to the one in Hilbert spaces.

In Section 3, we introduce the cascading operators in the setting of CAT(0) spaces and, following the reasoning by Lennard et al. in [1], we distinguish this family from other collections of operators, which encompass the most common generalizations of *asymptotically non-expansive mappings*, studied in metric spaces.

In Section 4, mainly inspired by the results by Dhompongsa et al. in [3] about asymptotically non-expansive mappings and Khamsi et al. in [4] concerning *asymptotically pointwise non-expansive mappings*, we establish a *demiclosedness principle* for cascading non-expansive mappings in CAT(0) spaces and derive some fixed point results for this family of operators. We also prove a Δ-*convergence theorem* for a Mann iteration process ([5]) using a cascading operator.

## 2. Preliminaries

A *geodesic* joining two points $x, y$ in a metric space $(X, d)$ is a mapping $\gamma : [0, 1] \to X$ such that $\gamma(0) = x$, $\gamma(1) = y$ and, for any $t, t' \in [0, 1]$, we have that:

$$d(\gamma(t), \gamma(t')) = |t - t'| d(x, y).$$

A metric space $(X, d)$ is *geodesic* if every two points in $X$ are joined by a geodesic. $(X, d)$ is said to be *uniquely geodesic*, if, for every $x, y \in X$, there is exactly one geodesic joining $x$ and $y$ for each $x, y \in X$, which we denote by $[x, y]$. The point $\gamma(t)$ in $[x, y]$ is also denoted by $(1 - t)x \oplus ty$.

As a subclass of the uniquely geodesic spaces, we have the CAT(0) spaces which usually are considered as the nonlinear analogue of Hilbert spaces. These spaces were introduced by Aleksandrov in [6].

**Definition 1.** *Let $(X, d)$ be a uniquely geodesic metric space, $x, y, z \in X$. We say that $(X, d)$ is* CAT(0) *if*

$$d\left(\left(\frac{x}{2} \oplus \frac{y}{2}\right), z\right)^2 \leq \frac{1}{2}d(x, z)^2 + \frac{1}{2}d(y, z)^2 - \frac{1}{4}d(x, y)^2.$$

The inequality from above is known as the CN inequality of Bruhat and Tits ([7]).

The following result is very useful in order to perform calculations in CAT(0) spaces.

**Proposition 1.** *([7] Proposition 2.2) Let $X$ be a* CAT(0) *space. Then, for all $x, y, w \in X$ and $t \in [0, 1]$,*

$$d((1 - t)x \oplus ty, w) \leq (1 - t)d(x, w) + td(y, w).$$

**Definition 2.** *A subset $C$ of a uniquely geodesic space is convex if, for any $x, y \in C$, we have that $[x, y] \subset C$. If $K \subset X$, we define*

$$\overline{conv}(K) = \bigcap\{D \subset X : D \supset K, D \text{ is closed and convex}\}.$$

A class of mappings widely studied (see [8–10] among others) in the setting of metric fixed point theory is the class of asymptotically non-expansive mappings.

**Definition 3.** *Let $(X, d)$ be a metric space. A mapping $T : X \to X$ is said to be asymptotically non-expansive if there exists a sequence of positive numbers $(k_n)$, with $\lim_{n \to \infty} k_n = 1$, such that, for all $n \in \mathbb{N}$ y $x, y \in X$,*

$$d(T^n x, T^n y) \leq k_n d(x, y). \tag{1}$$

These functions were defined first in the context of normed spaces by Kirk in [8] and properly extend the collection of non-expansive mappings, that is, those functions $T : X \to X$, such that $d(Tx, Ty) \leq d(x, y)$.

Finally, given a metric space $X$, a nonempty set $C \subset X$ and a mapping $T : C \to C$,

$$\text{Fix}(T) = \{x \in C : Tx = x\}.$$

## 3. Cascading Non-Expansive Mappings

In this section, we introduce the notion of cascading non-expansive mappings in the setting of complete CAT(0) spaces and compare it to other types of functions that include the most common generalizations of asymptotically non-expansive mappings studied both in metric spaces and Banach spaces.

**Definition 4.** *Let $(X, d)$ be a complete* CAT(0) *space and $C \subset X$ a closed convex set.*

*Define $C_0 = C$, $C_1 = \overline{conv}(T(C))$,..., $C_n = \overline{conv}(T(C_{n-1}))$. If there exists $\{k_n\} \subset [1, \infty)$ with $k_n \to 1$ as $n \to \infty$ such that, for all $x, y \in C_n$*

$$d(Tx, Ty) \leq k_{n+1}d(x, y),$$

*we say that $T$ is a cascading non-expansive mapping.*

Next, we recall the notions of totally asymptotically non-expansive mapping, asymptotically pointwise non-expansive mapping and mapping of an asymptotically non-expansive type.

**Definition 5.** *[11] Let $(X, d)$ be a metric space. A mapping $T : X \to X$ is called totally asymptotically non-expansive if there are nonnegative real sequences $(k_n^{(1)})$ and $(k_n^{(2)})$ with $k_n^{(1)}, k_n^{(2)} \to 0$, as $n \to \infty$ and a strictly increasing and continuous function $\psi : \mathbb{R}^+ \to \mathbb{R}^+$ with $\psi(0) = 0$ such that:*

$$d(T^n x, T^n y) \leq d(x, y) + k_n^{(1)} \psi(d(x, y)) + k_n^{(2)} \quad n \in \mathbb{N}, x, y \in X.$$

**Remark 1.** *This definition unifies several generalizations of the asymptotically non-expansive mappings.*

*If $\psi(t) = t$, we get the nearly asymptotically non-expansive mappings ([12]).*

*If $\psi(t) = t$ and for all $n \in \mathbb{N}$, $k_n^{(1)} = 0$ and*

$$k_n^{(2)} = \max\left(0, \sup_{x, y \in X} d(T^n x, T^n y) - d(x, y)\right), \text{ we recover the asymptotically non-expansive}$$

*mappings in the intermediate sense ([13]).*

*If $\psi(t) = t$ and for all $n \in \mathbb{N}$, $k_n^{(2)} = 0$, we have the asymptotically non-expansive mappings.*

**Definition 6.** *[14] Let $(X, d)$ be a metric space. A mapping $T : X \to X$ is called asymptotically pointwise non-expansive if there exists a sequence of mappings $\alpha_n : X \to [0, \infty)$ such that, for every $x \in X$, $\limsup_n \alpha_n(x) \leq 1$, it is verified that:*

$$d(T^n x, T^n y) \leq \alpha_n(x) d(x, y), \quad n \in \mathbb{N}, x, y \in X.$$

**Definition 7.** *[15] Let $(X, d)$ be a metric space. A mapping $T : X \to X$ is said to be an asymptotically non-expansive type if, for every $x \in X$,*

$$\limsup_n (\sup\{d(T^n x, T^n y) - d(x, y)) : y \in X\}) \leq 0.$$

In [1], some examples are given that prove that the collections of cascading non-expansive mappings and asymptotically non-expansive mappings differ, in the sense that, in general, neither collection is contained in the other. Taking as reference these examples, we distinguish the collection of cascading non-expansive mappings from the respective classes of functions given in Definitions 5, 6, and 7 in the setting of CAT(0) spaces. We recall that a linear space $X$ is CAT(0) if and only if $X$ is pre-Hilbert ([7]) Proposition 1.14 p. 167.

**Example 1.** *[1] (Example 2.5) Let $X = (\mathbb{R}, |\cdot|)$ and $K = \left[0, 1/\sqrt{2}\right]$. Let $\mathbb{Q}$ denote the set of rational numbers and $\mathbb{I} = \mathbb{R}/\mathbb{Q}$ be the set of irrational numbers. Define $U : K \to K$ such that:*

$$Ux = \begin{cases} \min\left(\sqrt{2}x, \frac{1}{\sqrt{2}}\right), & \text{if } x \in \mathbb{Q} \cap K, \\ Ux = 0, & \text{if } x \in \mathbb{I}/K. \end{cases}$$

*If, for all $n \in \mathbb{N}$, $K_n = \overline{\text{conv}}(U(K_{n-1}))$, where $K_0 = K$, then $0, \frac{1}{2} \in K_n$, but*

$$\left| U(0) - U\left(\frac{1}{2}\right) \right| = \left| 0 - \frac{1}{\sqrt{2}} \right| = \sqrt{2} \left| 0 - \frac{1}{2} \right|.$$

*From this, we conclude that $U$ is not a cascading non-expansive mapping. Observe that, for all $n \geq 2$ and $x \in K$, $T^n x = 0$ and hence $T$ is totally asymptotically non-expansive, asymptotic pointwise non-expansive and of an asymptotically non-expansive type.*

*The example shows that the family of asymptotically non-expansive maps is not contained in the class cascading operators in CAT(0) spaces.*

**Example 2.** *In $l_2$, we consider the following norm:*

$$\|x\| = \left( \sum_{i=1}^{\infty} |\gamma_i x(i)|^2 \right)^{1/2}$$

*where $(\gamma_i) \subset (0,1)$ is a sequence such that $\gamma_i \to 1$, $\frac{\gamma_{i+1}}{\gamma_i}$ is decreasing. It is straightforward to see that $X = (l_2, \|\cdot\|)$ is a Hilbert space and hence $\mathrm{CAT}(0)$ ([7] Proposition 1.14 p. 167).*
    *Let $C = C_0 = \left\{ x \in B_{l_2} : x(i) \geq 0, \quad i \in \mathbb{N} \right\}$ and, for $n \geq 1$,*

$$C_n = \{ x \in C : x(i) = 0, \quad 1 \leq i \leq n \}.$$

*Let $T : C \to C$ be such that*

$$(Tx)(j) = \begin{cases} 0, & \text{if } j = 1 \\ x(j-1), & \text{if } j > 1. \end{cases}$$

*It is easy to check that, for every $n \in \mathbb{N}$, $C_n = \overline{\mathrm{conv}}(T(C_{n-1}))$.*
    *Let us see that $T$ is a cascading non-expansive mapping. If $x, y \in C_n$, then*

$$\begin{aligned}
\|Tx - Ty\| &= \left( \sum_{j=1}^{\infty} (\gamma_{n+j+1})^2 |x(n+j) - y(n+j)|^2 \right)^{1/2} \\
&= \left( \sum_{j=1}^{\infty} \left( \frac{\gamma_{n+j+1}}{\gamma_{n+j}} \right)^2 (\gamma_{n+j})^2 |x(n+j) - y(n+j)|^2 \right)^{1/2} \\
&\leq \frac{\gamma_{n+2}}{\gamma_{n+1}} \|x - y\|.
\end{aligned}$$

*If $k_n = \frac{\gamma_{n+2}}{\gamma_{n+1}}$, then $k_n \to 1$; therefore, $T$ is a cascading non-expansive mapping. However,*

$$\|T^n e_1 - T^n e_2\| = \left( \gamma_n^2 + \gamma_{n+1}^2 \right)^{1/2} = \left( \frac{\gamma_n^2 + \gamma_{n+1}^2}{\gamma_1^2 + \gamma_2^2} \right)^{1/2} \|e_1 - e_2\|$$

*and, as $\left( \frac{\gamma_n^2 + \gamma_{n+1}^2}{\gamma_1^2 + \gamma_2^2} \right)^{1/2} \to \left( \frac{2}{\gamma_1^2 + \gamma_2^2} \right)^{1/2} > 1$, we deduce that $T$ is neither totally asymptotically non-expansive, pointwise asymptotically non-expansive nor of an asymptotically non-expansive type.*
    *This example shows that the family of cascading operators is not contained in the family of asymptotically non-expansive maps in $\mathrm{CAT}(0)$ spaces.*

**Example 3.** *Consider the following equivalence relation over the set $\mathbb{N} \times [0,1]$:*

$$(n,0)\mathcal{R}(m,0) \quad \text{and} \quad (n,t)\mathcal{R}(n,t) \quad n,m \in \mathbb{N}, t \in [0,1].$$

*Now, we define a metric on $X = (\mathbb{N} \times [0,1])/\mathcal{R}$ as:*

$$d((n,t),(n,s)) = |t - s| \quad \text{and if } n \neq m, \quad d((n,t),(m,s)) = t + s.$$

*It is easy to see that $(X, d)$ is a complete $\mathbb{R}$-tree ([7] p. 167), and, consequently, it is a complete $\mathrm{CAT}(0)$ space.*
    *Let $C_0 = X$ and, for every $n \in \mathbb{N}$, $C_n = ((\mathbb{N} - \{1, \dots, n\}) \times [0,1])/\mathcal{R}$.*

*Let $(\gamma_n) \subset [1, \infty)$ be such that $\gamma_n$ is a strictly decreasing sequence and $1 < \prod_{n=1}^{\infty} \gamma_n < \infty$. Define $T : C \to C$ such that, if $x = (m, t)$, then*

$$Tx = (m + 1, \min(1, \gamma_m t)).$$

*Observe that $T$ is well defined. Let us see that $T$ is a cascading non-expansive mapping. It can easily be checked that, for every $n \in \mathbb{N}$, $C_n = \overline{\text{conv}}(T(C_{n-1}))$.*
*Let $x, y \in C_n$ and consider the following cases:*
*Case 1. $x = (m, t)$, $y = (m, s)$ $(m \geq n + 1)$*

$$d(Tx, Ty) = |\min(1, \gamma_m t) - \min(1, \gamma_m s)| \leq \gamma_m |t - s| \leq \gamma_{n+1} d(x, y).$$

*Case 2. $x = (m, t)$, $y = (p, s)$ with $n + 1 \leq m < p$.*

$$d(Tx, Ty) = \min(1, \gamma_m t) + \min(1, \gamma_p s) \leq \gamma_m t + \gamma_p s \leq \gamma_{n+1} d(x, y).$$

*As $\gamma_n \to 1$ when $n \to \infty$, it follows that $T$ is cascading non-expansive. However, $T$ is not asymptotically non-expansive because, if $x = (1, t)$, $y = (1, s)$ with $s < t < \frac{1}{\prod_{n=1}^{\infty} \gamma_n}$, then:*

$$T^n x = (1 + n, \min(1, \gamma_n \ldots \gamma_1 t))$$
$$= (1 + n, \gamma_n \ldots \gamma_1 t)$$

*due to $t < \frac{1}{\prod_{n=1}^{\infty} \gamma_n}$.*
*Analogously, $T^n y = (1 + n, \gamma_n \ldots \gamma_1 s)$. Then,*

$$d(T^n x, T^n y) = \gamma_n \ldots \gamma_1 d(x, y).$$

*However, $\gamma_n \cdots \gamma_1 \to \prod_{n=1}^{\infty} \gamma_n > 1$ when $n \to \infty$. Consequently,*

$$\limsup_n d(T^n x, T^n y) > d(x, y)$$

*and $T$ does not belong to the families given in Definitions 5–7.*

**Remark 2.** *In [16], the author studied several generalizations of the asymptotically pointwise non-expansive mappings in the context of complete $\text{CAT}(0)$ spaces. However, throughout similar examples to those given above, it can be proved that the collection of cascading non-expansive mappings differs from such generalizations.*

## 4. Fixed Point Results for Cascading Operators

Cascading non-expansive operators constitute a new object of study in the framework of $\text{CAT}(0)$ spaces and, in general, they do not contain and are not contained in the collection of asymptotically non-expansive mappings as it was illustrated in Section 3. Thus, the theorems in this section are new and do not follow from the results related to asymptotically non-expansive maps.

Let $(X, d)$ be a complete $\text{CAT}(0)$ space, $(x_n)$ be a bounded sequence in $X$ and $x \in X$. Let $r(x, (x_n)) = \limsup_{n \to \infty} d(x, x_n)$. The asymptotic radius of $(x_n)$ is given by

$$r((x_n)) = \inf \{r(x, (x_n)) : x \in X\}$$

and the asymptotic center of $r((x_n))$ is the set

$$A((x_n)) = \{x \in X : r(x, (x_n)) = r((x_n))\}.$$

It is a well known fact that, in complete $\text{CAT}(0)$ spaces, $A((x_n))$ is a singleton ([17] Proposition 3.2).

The following notions of convergence were introduced by Lim and Kakavandi, re-spectively, in the setting of metric spaces. These notions resemble the weak convergence defined in Banach spaces and in fact they coincide with the weak convergence in Hilbert spaces ([18], p. 3452).

**Definition 8.** *Let* $(X,d)$ *be a complete* $\mathrm{CAT}(0)$ *space and* $(x_n)$ *be a bounded sequence in X.*

*(i) ([19] p. 180) We say that* $(x_n)$ $\Delta$-*converges to* $x \in X$ *if* $A\big((x_{n_k})\big) = \{x\}$ *for every subsequence* $(x_{n_k})$ *of* $(x_n)$.

*(ii) ([18]) We say that* $(x_n)$ *weakly converges to* $x \in X$ *if*

$$\lim_{n \to \infty} \big( d^2(x_n, x) - d^2(x_n, y) + d^2(x, y) \big) = 0, \quad y \in X.$$

**Remark 3.** *From Example 4.7 in [20], if follows that these notions of convergence are different.*

Let $(X, \|\cdot\|)$ be a Banach space, $C \subset X$ a nonempty closed convex set, and $T : C \to X$ be a mapping. If $I : X \to X$ denotes the identity map, it is said that $I - T$ is demiclosed at zero, if, for any sequence $(x_n) \subset C$ such that $x_n$ weakly converges to $x$ and $\|(I - T)(x_n)\| \to 0$, we have that $Tx = x$.

One of the fundamental results in metric fixed point theory for non-expansive map-pings is the demiclosedness principle of Browder [21], which establishes that, if $X$ is an uniformly convex Banach space, $C \subset X$ is a closed convex set and $T : C \to X$ is a non-expansive mapping, then $I - T$ is demiclosed.

Several works ([4,12,17,22,23] among others) have been devoted to prove demiclosed-ness principles both in Banach and metric spaces for mappings which generalize the non-expansive ones. The following theorem could be interpreted as a demiclosedness principle for cascading non-expansive mappings with respect to the convergence given in (*i*) in Definition 8.

**Theorem 1.** *Let* $(X, d)$ *be a complete* $\mathrm{CAT}(0)$ *space and* $C$ *a closed convex subset of X. Let* $T : C \to C$ *be a cascading non-expansive mapping and* $(k_n)$ *be given as in Definition 4 with* $\prod_{j=1}^{\infty} k_j < \infty$. *If* $(x_n) \subset C$ $\Delta$-*converges to* $w$ *and* $\lim_{n \to \infty} d(x_n, Tx_n) = 0$, *then* $Tw = w$.

**Proof.** Since $d(x_n, Tx_n) \to 0$, there exists a subsequence $(x_{n_l})$ such that $d(x_{n_l}, T^l x_{n_l}) \to 0$ whenever $l \to \infty$. Let us see that, if $y_l = T^l x_{n_l}$, then $y_l$ $\Delta$-converges to $w$. Indeed, as $(x_n)$ $\Delta$-converges to $w$, for any $x \in X$:

$$\limsup_{j \to \infty} d\big(y_{l_j}, w\big) = \limsup_{j \to \infty} d\big(x_{n_{l_j}}, w\big) \le \limsup_{j \to \infty} d\big(x_{n_{l_j}}, x\big)$$
$$= \limsup_{j \to \infty} d\big(y_{l_j}, x\big).$$

From this, we conclude that $w \in A\big(\{y_{l_j}\}\big)$, but, since Proposition 3.2 in [17] im-plies that $A\big(\{y_{l_j}\}\big)$ is a singleton, it follows that $A\big(\{y_{l_j}\}\big) = \{w\}$ and therefore $(y_l)$ $\Delta$-converges to $w$. By Proposition 3.2 in [24], $w \in D = \cap_{n=1}^{\infty} C_n$. Observe that

$$d(Ty_l, y_l) = d\big(T^{l+1} x_{n_l}, T^l x_{n_l}\big) \le \left( \prod_{i=1}^{l} k_i \right) d\big(Tx_{n_l}, x_{n_l}\big) \to 0$$

when $l \to \infty$.

Hence, for $z \in X$, $r(z, (Ty_l)) = r(z, (y_l))$.

In particular,

$$r(Tw, (y_l)) \quad = \quad \limsup_{l \to \infty} d(Tw, Ty_l) \leq \limsup_{l \to \infty} k_l d(w, (y_l))$$
$$= \quad r(w, (y_l)),$$

but, since $(y_l)$ $\Delta$-converges to $w$, $r(w, (y_l)) \leq r(Tw, (y_l))$.

Consequently, $r(w, (y_l)) = r(Tw, (y_l))$ and, since $A((y_l))$ is a singleton, we get that $w = Tw$. $\quad \square$

Theorem 1 also holds when we consider the notion of convergence given in ii) in Definition 8.

**Corollary 1.** *Let* $(X, d)$ *be a complete* CAT$(0)$ *space and* $C$ *a closed convex subset of* $X$. *Let* $T : C \to C$ *be a cascading non-expansive mapping and* $(k_n)$ *be given as in Definition 4 with* $\prod_{j=1}^{\infty} k_j < \infty$. *If* $(x_n) \subset C$ *is such that* $\lim_{n \to \infty} d(x_n, Tx_n) = 0$ *and* $(x_n)$ *weakly converges to* $w$, *then* $Tw = w$.

**Proof.** If $(x_n)$ weakly converges to $w$, Proposition 2.5 in [20] implies that $(x_n)$ $\Delta$-converges to $w$, and the conclusion follows from Theorem 1. $\quad \square$

By considering the hypothesis of boundedness over $C$, we have that:

**Corollary 2.** *Let* $(X, d)$ *be a complete* CAT$(0)$ *space and* $C$ *a closed convex subset of* $X$. *Let* $T : C \to C$ *be a cascading non-expansive mapping and* $(k_n)$ *be given as in Definition 4 with* $\prod_{j=1}^{\infty} k_j < \infty$. *If* $C$ *is bounded, the set of fixed points of* $T$, *denoted by* Fix$(T)$, *is a nonempty closed convex set.*

**Proof.** Let $x_0 \in C$ and $x_n = T^n x_0$. From [24] (p. 3690), $(x_n)$ has a subsequence $(x_{n_j})$ which $\Delta$-converges to $w$ and by Proposition 3.2 in [24]

$$w \in \cap_{j=1}^{\infty} \overline{\text{conv}} \left\{ x_{n_j}, x_{n_{j+1}}, \dots \right\} \subset \cap_{n=1}^{\infty} C_n.$$

Consequently, $\cap_{n=1}^{\infty} C_n$ is a nonempty set, and, since $T : \cap_{n=1}^{\infty} C_n \to \cap_{n=1}^{\infty} C_n$ is non-expansive, the conclusions follows from Theorem 5.1 in [4]. $\quad \square$

**Lemma 1.** *Let* $(X, d)$ *be a complete* CAT$(0)$ *space and* $C$ *a closed convex bounded subset of* $X$. *Let* $T : C \to C$ *be a cascading non-expansive mapping and* $(k_n)$ *be as in Definition 4 with* $\prod_{j=1}^{\infty} k_j < \infty$. *Consider the following variant of the Mann iteration process ([5]):*

$$x_{n+1} = (1 - \alpha_n) Tx_n \oplus \alpha_n T^2 x_n \tag{2}$$

*where* $x_0$ *is any element in* $C$ *and there exist* $\beta_1, \beta_2 > 0$, *such that, for all* $n \in \mathbb{N}$, $0 < \beta_1 \leq \alpha_n \leq 1 - \beta_2 < 1$. *It holds that:*

1. *If* $w \in$ Fix$(T)$, *then* $\lim_{n \to \infty} d(x_n, w)$ *exists.*
2. $\lim_{n \to \infty} d(x_n, Tx_n) = 0$.

**Proof.** Remember that, by Corollary 2, Fix$(T)$ is a nonempty set. Let $w \in$ Fix$(T)$.

1. By Proposition 1,

$$d(x_{n+1}, w) \quad = \quad d((1 - \alpha_n) Tx_n \oplus \alpha_n T^2 x_n, w) \leq (1 - \alpha_n) d(Tx_n, w)$$
$$+ \quad \alpha_n d(T^2 x_n, T^2 w)$$
$$\leq \quad (1 - \alpha_n) k_n d(x_n, w) + k_{n+1} k_n \alpha_n d(x_n, w)$$
$$= \quad (1 + \alpha_n k_n (k_{n+1} - 1)) d(x_n, w)$$

and, from Lemma 1.2 in [25], we get that $\lim_{n\to\infty} d(x_n, w)$ exists.

2. Let $r = \lim_{n\to\infty} d(x_n, w)$. Since $x_n \in C_n$ and $w \in \text{Fix}(T)$,

$$\limsup_{n\to\infty} d(Tx_n, w) = \limsup_{n\to\infty} d(Tx_n, Tw) \le \limsup_{n\to\infty} k_{n+1} d(x_n, w) = r.$$

Similarly, $\limsup_{n\to\infty} d(w, T^2 x_n) \le r$, so, by Lemma 4.5 in [17], we have that $\lim_{n\to\infty} d(Tx_n, T^2 x_n) = 0$.

On the other hand,

$$\begin{aligned} d(x_{n+1}, Tx_{n+1}) &\le d\left(x_{n+1}, T^2 x_n\right) + d\left(T^2 x_n, Tx_{n+1}\right) \\ &\le (1-\alpha_n)d(Tx_n, T^2 x_n) + k_n\alpha_n d(Tx_n, T^2 x_n) \\ &\le (1 - \alpha_n + k_n\alpha_n)d\left(Tx_n, T^2 x_n\right) \to 0. \end{aligned}$$

□

The following example shows a simple application of Theorem 1 and generalizes Example 3.

**Example 4.** *Let $X$ be the space described in Example 3. For simplicity, we write $(m, t)$ to represent the class $[(m, t)]$ if $t > 0$ and define $w_0 = [(m, 0)]$. Let $T : X \to X$ be a cascading operator for which the sequence $(k_n)$ is given as in Definition 4, $\prod_{n=1}^{\infty} k_n < \infty$, and there exists $(m_0, t_0) \in X$, such that for all $N \in \mathbb{N}$,*

$$\{T^n(m_0, t_0) : n \in \mathbb{N}\} \cap \{(m, t) : m \ge N, 0 < t \le 1\} \ne \varnothing. \tag{3}$$

*(Example 3 shows that such $T$ exists) Then, $w_0 = (m, 0) \in \text{Fix}(T)$.*

*Let $(m_0, t_0)$ be the point for $T$ satisfying (3). If $x_0 = (m_0, t_0)$ and $x_{n+1} = \frac{1}{2}Tx_n \oplus \frac{1}{2}T^2 x_n$, Lemma 1 implies that $d(Tx_n, x_n) \to 0$. Define $\pi_1 : X \to \mathbb{N}$ and $\pi_2 : X \to [0, 1]$ as*

$$\pi_1((m, t)) = \begin{cases} m, & \text{if } t > 0 \\ 1, & \text{if } t = 0. \end{cases}$$

*and $\pi_2((m, t))) = t$. From condition (3), by passing to a subsequence if necessary, we may assume that, for all $j \in \mathbb{N}$, $\pi_1\left(x_{n_j}\right) \ne \pi_1\left(Tx_{n_j}\right)$. Thus, $d\left(x_{n_j}, Tx_{n_j}\right) = \pi_2 x_{n_j} + \pi_2 Tx_{n_j}$ and $\lim_{j\to\infty} d\left(x_{n_j}, w_0\right) = 0$. Since convergence in metric implies $\Delta$-convergence, $x_{n_j}$ $\Delta$-converges to $w_0$ and, from Theorem 1, it follows that $Tw_0 = w_0$.*

**Lemma 2.** *Let $(X, d)$ be a complete $\text{CAT}(0)$ space and $T : C \to C$ be a cascading non-expansive mapping with $(k_n)$ as in Definition 4 and $\prod_{n=1}^{\infty} k_n < \infty$. Let $(x_n)$ be a sequence in $C$ such that $\lim_{n\to\infty} d(x_n, Tx_n) = 0$ and $(d(x_n, w))$ converges for all $w \in \text{Fix}(\text{T})$. Then, $(x_n)$ $\Delta$-converges to a fixed point of $T$.*

**Proof.** It is similar to the proof of Lemma 2.10 in [3]. □

Finally, from Lemmas 1 and 2, we conclude that the Mann iteration process defined in Equation (2), $\Delta$-converges to a fixed point of $T$.

**Theorem 2.** *Suppose that $C$ is a closed convex bounded subset of $(X, d)$ and let $T : C \to C$ be a cascading non-expansive mapping with $(k_n)$ as in Definition 4 and $\prod_{n=1}^{\infty} k_n < \infty$. Let $x_0$ be any initial point in $C$ and $(x_n)$ the sequence defined in Equation (2). Then, $(x_n)$ $\Delta$-converges to a fixed point of $T$.*

**Proof.** By Lemma 1, $d(x_n, Tx_n) \to 0$ when $n \to \infty$ and, for any $w \in \text{Fix}(T)$, $\lim_{n\to\infty} d(x_n, w)$ exists. Therefore, Lemma 2 implies that $(x_n)$ $\Delta$-converges to a fixed point $w_0$ of $T$. $\square$

The theorems introduced in Section 4 are inspired by some well known results previously studied in CAT(0) spaces for asymptotically non-expansive maps. It would be interesting to find general fixed point theorems which include both families of maps and to determine conditions under which the two families coincide.

**Author Contributions:** Both authors contributed equally in the development of this work. Both authors have read and agreed to the published version of the manuscript.

**Funding:** This research was partially supported by Marie Curie Skłodowska University, 20-031 Lublin, Poland.

**Institutional Review Board Statement:** Not applicable.

**Informed Consent Statement:** Informed consent was obtained from all subjects involved in the study.

**Data Availability Statement:** Data sharing not applicable No new data were created or analyzed in this study. Data sharing is not applicable to this article.

**Acknowledgments:** We would like to thank the referees for their helpful comments.

**Conflicts of Interest:** The authors declare no conflict of interest.

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
