# Peer review of "Cascading Operators in CAT(0) Spaces"

_axioms, doi:10.3390/axioms10010020_

Round 1

Reviewer 1 Report

This paper contains abstract, introduction (the motivation for research), cascading nonexpansive mappings (known and new definitions of the notion in CAT(0) spaces, known and new examples), fixed point results (1 new definition, 1 new example, 2 new theorems, 2 new lemmas, 2 new corollaries) and references (25 items). 

Upon reviewing the paper I have made the following remarks:

  • in whole paper it should write "+∞" instead of "∞", "limn→+∞" instead of "limn→∞", "∑+∞" instead of "∑", "∏+∞" instead of "∏", "∩+∞" instead of "∩";
  • check again the example 3 (line 4).

General opinion

This paper contains several new and original results. All the proofs seem to be correct. This paper also contains several examples which support the theoretical results. The list of references should include some new recent items for this field of study, such as:

  • Vesna Todorčević, Harmonic Quasiconformal Mappings and Hyper-
    bolic Type Metrics, Springer Nature Switzerland, AG 2019,
  • Radisa Jovanović et al., Ensemble of various neural networks for pre-
    diction of heating energy consumption, Energy and Buildings, VOL 94, pp.189-199, doi:10.1016/j.enbuild.2015.02.052,

  • V. S. Chary et al., Some fixed point theorems for modified JS—G-
    contractions and an application to integral equations, J. Appl. Math. and Informatics Vol. 38 (2020), No. 5-6, pp. 507-518.

Reviewer 2 Report

publish as stands

Reviewer 3 Report

Definition 4: The left-hand-side has apparently  the superscript "n" mixed in the mapping T.

The section 3 does not contain non-trivial illustrative examples. Some of such examples should be incorporated and discussed.

It can be of interest to formally discuss under which conditions the set of cascading non-expansive operators is  either included or identical to the set of asymptotically  non-expansive ones and vice-versa.

Round 2

Reviewer 1 Report

Upon reviewing the paper I have made the following remarks:

  • in whole paper it should write "+∞" instead of "∞", "limn→+∞" instead of "limn→∞", "∑+∞" instead of "∑", "∏+∞" instead of "∏", "∩+∞" instead of "∩".

The list of references should include some new recent items for this field of study, such as:

  • Vesna Todorčević, Harmonic Quasiconformal Mappings and Hyper-
    bolic Type Metrics, Springer Nature Switzerland, AG 2019,
  • Radisa Jovanović et al., Ensemble of various neural networks for pre-
    diction of heating energy consumption, Energy and Buildings, VOL 94, pp.189-199, doi:10.1016/j.enbuild.2015.02.052,

  • V. S. Chary et al., Some fixed point theorems for modified JS—G-
    contractions and an application to integral equations, J. Appl. Math. and Informatics Vol. 38 (2020), No. 5-6, pp. 507-518.

Reviewer 3 Report

The paper is improved and it is well- formalized and discussed with examples. It  contains some new results in the field and it is also interesting for some of the readers of this journal.